# On the Factors Governing Austenite Stability: Intrinsic versus Extrinsic

**DOI:** 10.3390/ma13153440

**Published:** 2020-08-04

**Authors:** Binbin He

**Affiliations:** Department of Mechanical and Energy Engineering, Southern University of Science and Technology, Shenzhen 518055, China; hebb@sustech.edu.cn; Tel.: +86-755-88015374

**Keywords:** austenite stability, martensitic transformation, TRIP effect, defects, advanced high strength steels

## Abstract

In this review, we separate the different governing factors on austenite stability into intrinsic and extrinsic factors, depending on the domain defined by austenite grain boundaries. The different measuring techniques on the effectiveness of the governing factors in affecting the austenite stability are discussed. On the basis of the austenite stability, a new alloy design strategy that involves the competition between the intrinsic and extrinsic factors to control the transformation-induced plasticity (TRIP) effect to realize the stronger the more ductile steel is proposed. The present review may provide new insights into the development of novel thermal-mechanical processing to advance the mechanical properties of steels for industrial applications.

## 1. Introduction

Steels have been the working horse for the automotive industry since the 1920s. However, their share in automobiles is decreasing due to the competition from other materials, including Al alloys [1] and even wood [2]. The specific strength of steels may not be as competitive as other structural materials owing to its relatively high density (~7.8 kg/cm^3^). Although the density of steels can be effectively reduced by alloying of the lightweight element such as Al, it easily reaches a limit (~13 wt.%) beyond which undesirable brittle intermetallic compound is formed [3]. Nevertheless, the specific strength of steels can be largely increased by elevating their strength through engineering defects, such as boundaries (grain, twin, and interface) [4,5,6], dislocations [7], and precipitations [8]. However, the introduction of defects frequently deteriorates the ductility, which is known as the trade-off between strength and ductility [9].

Steels can be made strong and ductile by properly utilizing the transformation-induced plasticity (TRIP) effect [10,11]. Moreover, the toughness and formability of steels can also be improved with the assistance of the TRIP effect [12,13,14,15]. This TRIP effect plays a pivotal role in developing the advanced high strength steels (AHSSs) [16], including the TRIP-assisted steel [17], maraging TRIP steel [18], medium Mn TRIP steels [19,20], quenching and partitioning (Q&P) steels [21,22], and carbide free bainite (CFB) steels [23]. Austenite is the common phase in the above AHSS and is the source of the TRIP effect. The retained austenite is metastable and could transform to martensite during plastic deformation, leading to the operation of the TRIP effect to enhance the work hardening behavior. Although the hardening induced by the TRIP effect is obvious, the underlying mechanism is still under debate. Generally, there are three possible explanations for the TRIP effect, including the generation of new dislocations [24], dynamic strain partitioning [25], and localized stress relaxation [26]. It is reported that the direct contribution of the TRIP effect to the plasticity of steels is expected to be small [27], which may suggest that the timing for the operation of the TRIP effect is pivotal for mechanical properties of steels containing austenite. Both exhaustion of the TRIP effect at small strain and suppression of the TRIP effect at large strain are undesirable for the mechanical properties. The timing for the operation of the TRIP effect is governed by the mechanical stability of austenite grains.

An individual austenite grain is separated from other counterparts by its grain boundaries. The austenite grain interior is distributed with various defects including the interstitial/substitutional atoms, dislocations, and possibly twin boundaries (Figure 1). The above defects including the grain boundaries affect the austenite stability. These defects are inherent to the single austenite grain, intrinsically affecting its stability, and thus could be termed as the intrinsic factors. In contrast, the adjacent grains may also affect the stability of interested austenite grain via stress/strain partitioning, thus the effect associated with the adjacent grains is termed as the extrinsic factors (Figure 1).

The structure of the present review is arranged as below. Firstly, the governing factors on the austenite stability are thoroughly reviewed from the aspect of intrinsic and extrinsic based on their physical domain. Secondly, the different techniques to evaluate the effectiveness of these governing factors in controlling the austenite stability are discussed. Then, the alloy design strategies are discussed based on the intrinsic and extrinsic factors. It is proposed that the competition between the intrinsic and extrinsic factors may lead to the development of high-performance steels for automotive applications. Finally, a prospect for the unresolved problems concerning with the austenite stability is discussed.

## 2. The Factors Govern Austenite Stability

The austenite stability can be separated into the thermal stability and mechanical stability, defined based on the external driving force (cooling or stress). Here, thermal stability and mechanical stability of austenite are not differentiated because they can be equivalent to each other on the aspect of energy. Moreover, both thermal and mechanical stability can be affected by intrinsic and extrinsic factors. The extrinsic factors such as the stress from adjacent grains owing to varied thermal expansion ratio may affect the thermal stability of austenite during the cooling process [28]. The intrinsic factors are related to the austenite grain domain defined by its grain boundaries, including interstitial/substitutional atoms (chemical composition), dislocations, and grain boundaries (grain size/morphology) (Figure 1). The extrinsic factors are those applied by adjacent grains, involving the stress/strain partitioning, strength of the matrix, stress state (i.e., hydrostatic pressure), grain orientation, and strain rate (i.e., adiabatic heating) (Figure 1).

### 2.1. Intrinsic Factors

#### 2.1.1. Chemical Compositions

The alloying elements could occupy the interstitial or substitutional positions of the steels depending on its size. Any element that increases the change of chemical free energy among austenite and martensite or expands the austenite regime in phase diagram tends to stabilize the austenite [29]. The stabilization of austenite can be achieved by alloying the nitrogen (N), carbon (C), manganese (Mn), and nickel (Ni). These elements enlarge the difference of chemical free energy (Δ*G*) between the parent austenite (*γ*) and product martensite (*α*′), which can be estimated as below [30]:(1)$$\Delta G^{\gamma \to \alpha^{\prime }}=\left(1-x\right)\Delta G_{Fe}^{\gamma \to \alpha^{\prime }}+x\Delta H^{\gamma \to \alpha^{\prime }}$$
where *x* is the element contents (in atom%) and Δ*H* is the change of enthalpy. The martensitic transformation occurs at a temperature that is well below the *T*_0_ temperature (a temperature where the change of free energy between austenite and martensite is zero), which is owing to the presence of an energy barrier contributed by the interfacial energy and the strain energy. The effect of chemical compositions on the austenite stability can be evaluated from the martensite start (*M_s_*) temperature. The austenite grains with a higher *M_s_* temperature indicate their lower stability. Many formulae have been developed to predict the *M_s_* temperature [31,32,33,34], with a classical one provided by Andrew as below [35]:(2)$$\begin{array}{c} M_{s}{(}^{{}^{\circ}}C)=539-423\left(\%C\right)-30.4\left(\%\mathrm{Mn}\right)-17.7\left(\%\mathrm{Ni}\right)-12.1\left(\%\mathrm{Cr}\right)\\ -7.5\left(\%\mathrm{Mo}\right) \end{array}$$
where the element contents are in wt.% in Equation (2). It is empirically determined based on a very large data set [35].

The linear dependence of *M_s_* temperature on the C content, as shown in Equation (2), can be confirmed from the measurement of *M_s_* temperature in Q&P steels with varied C content [36]. The precipitation of cementite in austenite during tempering will destabilize the austenite in bainitic steel, which is indirect evidence for the effect of C on the austenite stability [37]. Although the C atoms are widely adopted in steels to stabilize the austenite, it is found that the stability of austenite in 18% Cr-8% Ni steel containing N is higher than that comprising C [38]. Therefore, N is believed to be the strongest element to stabilize the austenite. However, the difficulty of nitriding makes the N less favorable as compared with the C. The Ni element, which is a key element in the design of maraging steel [39], stainless steel [40], and Invar alloy [41], also tends to stabilize the austenite grains. However, the Ni element is not cost-effective as compared with the Mn. Moreover, it is less effective in retaining austenite as compared with the Mn, according to Equation (2). Note that the prediction of *M_s_* temperature using Equation (2) is confident for Mn content as high as 10 wt.% [42]. The Mn and C are widely used in the design of the third generation of AHSS [43,44]. To fully utilize the potential of Mn and C in stabilizing the austenite grains, the element partitioning of these two elements is enabled by adopting different thermal processing such as intercritical annealing, quenching and partitioning, and bainitic holding. The effect of Al on the *M_s_* temperature is not shown in Equation (2), but it can be included with a factor of +30 °C/wt.% [45]. This is different from the finding that the addition of Al increases the austenite stability in TRIP-assisted steels [46]. Such a discrepancy can be reconciled based on the fact that the alloying of Al inhibits the cementite precipitation during the bainitic transformation, and thus more C content is partitioned into the austenite grain. Similarly, the phosphorus stabilizes the individual austenite grains in the same way [46]. Although hydrogen embrittlement is frequently observed for AHSS, the hydrogen itself is found to hardly affect the austenite stability during tensile deformation [47]. The effect of Cr on the *M_s_* temperature is not shown in Equation (2); it is suggested that the addition of Cr decreases the *M_s_* temperature with a factor of 12.1 °C/wt.% [34]. In other words, the alloying of Cr tends to enhance the austenite stability, which is supported by the experimental work on high carbon steel [48].

#### 2.1.2. Dislocations

Dislocation is a line defect in crystalline solids. It plays an important role in nucleation and growth of martensite and, therefore, it largely affects the austenite stability in steels. However, the effect of dislocations on the austenite stability is relatively complex and controversial. The dissociation of dislocations into partial dislocations and the intersecting shear owing to two arrays of partial dislocations could lead to a body-centered cubic (bcc) structure that may serve as the martensite embryo [49]. The dislocation loops generated at the embryo/austenite interface may glide to extend the interface and facilitate the growth of embryo [30]. In other words, the dislocations could assist the martensitic transformation on the aspects of nucleation and growth. Nevertheless, the mechanical stabilization induced by the presence of dislocation debris is reported for austenite grains with large plastic deformation [50]. In other words, the dislocations could act as barriers to the growth of martensite [51,52]. The critical strain for mechanical stabilization induced by dislocations can be derived by equating the chemical driving force (Δ*G*) to the force resisting the glide of the interface, as below [51]:(3)$$b\Delta G=\frac{1}{8\pi \left(1-\nu \right)}Gb^{3/2}{\left(\frac{\varepsilon }{L}\right)}^{1/2}+\tau_{S}b$$
where *b* is Burgers vectors, *G* is the shear modulus and *ν* is the Poisson’s ratio, *ε* is the plastic strain, *L* is the mean free distance, and *τ_s_* is the equivalent shear stress. The above equation can be used to calculate the critical strain for the mechanical stabilization in stainless steel and to explain the localized mechanical stabilization during bainitic transformation [51]. Note that the controversial role of dislocations on the austenite stability may arise from the different amounts of plastic deformation.

The complex role of dislocations on the austenite stability is rationalized based on the amounts of dislocations [53,54]. The presence of a small amount of dislocations in austenite grain facilitates the nucleation of martensite by initiating the plastic accommodation of transformation strain of martensite, while the introduction of a large amount of dislocations suppresses the nucleation owing to the higher strength of austenite [53]. However, such an explanation does not consider the general plastic deformation mechanism. A recent work reports that the small deformation increases the *M_s_* temperature, while large deformation decreases the *M_s_* temperature, which is explained by combining general plastic deformation and martensitic transformation mechanisms [55]. The small deformation involves the generation of geometrically necessary dislocations (GND) close to the austenite grain boundaries where the potent martensite embryo exists [56]. The presence of GND adjacent to the austenite grain boundary may enhance the potency of the martensite embryo. The sparsely distributed dislocation at the interior of austenite grain after small deformation is not able to suppress the growth of the martensite lath (Figure 2a–c). Consequently, the *M_s_* temperature is increased after a small deformation. The large deformation results in the formation of intensive dislocations at the interior of austenite grain and the arrangement of these dislocations leads to the formation of subgrains. The presence of sub-grains substantially inhibits the growth of martensite lath (Figure 2d,e), refining the martensite block size and thus decreasing the *M_s_* temperature after large deformation [55]. It is reported that the austenite grains with prior deformation are less stable than those without deformation, suggesting that the plastic deformation may destabilize the austenite grains [57]. Note that the amount of prior deformation is 2%, which could be considered as the small deformation. Thus, the above destabilization of austenite after small deformation is consistent with the increase of *Ms* temperature after small deformation [55].

#### 2.1.3. Grain Size

In general, the small austenite grains are more stable than the large austenite grains during either quenching [58,59,60,61,62] or deformation processes [63]. Thus, grain refinement has been employed to retain the austenite grains at room temperature [43,44]. Note that the grain refinement hardly affects the lath width of martensite [64]. Considering the lath width of martensite in low C steel (100 nm to 500 nm) [65], it is reasonable to expect that the grain refinement plays a vital role in stabilizing the ultrafine austenite grain (<0.5 μm). This is confirmed from the experimental work on medium Mn steel, where the optimal austenite grain size is 0.5–0.6 μm [63,66]. The grain size dependent austenite stability can be interpreted from the aspect of *M_s_* temperature through the following empirical equation [58]:(4)$$M_{s}{(}^{{}^{\circ}}C)=Ms_{0}-Ax_{C}-BV_{\gamma }^{-\frac{1}{3}}$$
where *V_γ_* represents the volume of austenite grain. *M_s0_* is the temperature determined by the chemical compositions, besides C content. *A* = 423 wt.%/°C according to Andrew’s equation [35]. *B* is a fitting parameter that can be determined through the experimental data. According to Equation (4), the smaller grain size leads to a higher *M_s_* temperature, which can be verified in steels with different chemical compositions (Figure 3).

However, it is reported the prediction based on this model underestimates the effect of grain size in ultrafine austenite grains (<0.5 μm) [66]. Note that the pairing of the martensite variants is different for austenite grain with varied sizes. It is reported that 24 martensite variants can be found in the large austenite grains, while only several dominated variants can be observed in the small austenite grains [68]. The combination of the six martensite variants in a packet can substantially reduce the overall transformation strain [69]. In contrast, the formation of single or limited martensite variants leads to very high transformation strain, and thus a high nucleation strain energy barrier [69]. Therefore, a significant stabilization of austenite could be observed for critical austenite grain size [68]. The dependence of *M_s_* temperature on grain size can be modeled from a thermodynamic aspect with consideration of the strength of austenite and the stored energy, both of which are related to the austenite grain size [70]. It is reported that this thermodynamic model can predict the effect of both chemical compositions and grain size on the *M_s_* temperature and is more accurate than the empirical models [70].

Despite the different grain size effect, that is, the smaller, the less stable, that could be observed during either quenching [71] or deformation processes [72,73], it is generally induced by the other factors such as segregation of elements and the defects. For instance, the increased *M_s_* temperature at high austenitization temperature (corresponding to large grain size [74]) in Al-alloyed steels is induced by the grain boundary segregation of Al [71]. The small austenite grain is less stable than the large one in stainless steel owing to the precipitations of CrN in small grains at a low annealing temperature [73]. The smaller being less stable observed in maraging TRIP steel can be ascribed to the formation of deformation twins in large austenite grains [72].

#### 2.1.4. Morphology

The austenite grains in AHSS generally have two types of morphology, including film/lath and blocky/globular (Figure 4). These different morphologies have resulted from either the single heterogeneous transformation such as martensitic transformation [75] and bainitic transformation [76,77,78], or the combination of different phase transformations [22]. The reversed austenite grains could have lath morphology and granular morphology depending on the rolling state [79], growth mode [80], and annealing conditions (Figure 4). In addition to the morphology, the austenite grains obtained by phase transformation could have different C content and dislocation density. For instance, the austenite with blocky type found between the ferrite grains has a higher C content than the austenite with film type adjacent to the lath martensite [22].

The dependence of austenite stability on its morphology is frequently reported in the literature for different steel grades [81,82,83,84,85]. It is generally found that the blocky austenite is less stable as compared with the filmy austenite during plastic deformation [22,86]. Nevertheless, the effect of morphology on the austenite stability generally involves the contribution from the other factors. The austenite grains with different morphology have varied microstructural features including the chemical compositions, grain size (volume), and adjacent phases. Therefore, it is relatively complex to discuss the influence of morphology on the austenite stability in steels with a multi-phase microstructure. For instance, it is found that the blocky austenite adjacent to ferrite has a higher C content than the filmy austenite close to the lath martensite in Q&P steel [22]. The C atoms are the strong austenite stabilizer. However, the blocky austenite transforms earlier than the filmy austenite [22]. The early transformation of blocky austenite as compared with the filmy austenite is ascribed to the lack of both hydrostatic pressure from the adjacent martensite and the high yield stress of lath martensite (or shielding effect) [22]. The hydrostatic pressure stabilizes the austenite grains by suppressing the volume expansion during martensitic transformation [87]. The lath martensite with high yield strength cannot accommodate the transformation strain of product martensite as easily as compared with the ferrite with low yield strength, which is analogous to the shielding effect that the strong adjacent phase protects austenite from deformation [86,88]. Besides, it is expected that the filmy austenite grains have a higher dislocation density than the blocky austenite owing to the accommodation of transformation strain from adjacent lath martensite. Similarly, for steel produced by single displacive shear transformation, the blocky austenite should contain a lower dislocation density than the film austenite because the dislocations generated during shear transformation are generally localized at the interface [89]. The blocky austenite with a larger volume shall have a lower dislocation density. Therefore, the effect of morphology, which is supposed to intrinsically affect the austenite stability, is frequently influenced by the other governing factors, which can be attributed to the complex microstructures derived from phase transformation as well as the macroscopic tensile deformation, both of which make the separation of morphology from other governing factors difficult.

Most of the existing investigations on the effect of morphology on austenite stability fall in the dilemma that the intrinsic effect of morphology is overwhelmed by the other governing factors, including intrinsic (grain volume, chemical composition, dislocations) and extrinsic (stress state, the strength of the matrix) factors. Moreover, most of the explanations are based on an implicit assumption that the morphology of austenite does not affect the nucleation energy barrier of martensite. Although some studies suggest that the shielding effect is dominant as compared with the lamellar morphology, they fail to quantify the contribution of morphology to the austenite stability [88]. It has been demonstrated that the strain energy of martensitic transformation is inversely proportional to the aspect ratio of the martensite plate [90,91]. Thus, it is important to assess the effect of austenite morphology on the aspect ratio of martensite. As the martensite lath width is not changed by the austenite grain size [64], it is expected that the filmy austenite could lead to the formation of martensite with a low aspect ratio, which may intrinsically stabilize the austenite grains.

### 2.2. Extrinsic Factors

#### 2.2.1. Stress/Strain Partitioning

The third generation of AHSS generally consists of multiple phases with the same component of retained austenite. The other ferritic phases could be ferrite, bainite, or martensite depending on the steel grades. The difference in mechanical properties between the retained austenite and ferritic phases leads to the stress/strain partitioning among them during the plastic deformation, and consequently affects austenite stability. The austenite stability is closely related to the martensitic transformation. The martensitic transformation under applied stress can be further classified as the stress-induced/strain-induced martensitic transformation depending on the value of the applied stress in comparison with the yield strength (Figure 5a) [92]. The martensitic transformation that occurs before the dislocation glide at a stress lower than the yield stress is known as the stress-induced martensitic transformation, while the scenario in which the dislocation motion is ahead of the martensitic transformation under applied stress is considered as the strain-induced martensitic transformation (Figure 5a) [93]. In other words, the definition of stress-/strain-induced martensitic transformation depends on the race between dislocation glide and martensitic transformation in austenite. Therefore, the stress-induced martensitic transformation means that applied stress below the yield stress is sufficient to provide the required change of chemical free energy in terms of mechanical interaction energy (*G_mech_*) to make the martensitic transformation possible at an elevated temperature (Figure 5b) [87]. In other words, the applied stress facilitates the growth of pre-existing martensite embryo, which is similar to the quenching process. The strain-induced martensitic transformation involves the development of extra martensite embryos from the defects generated during the plastic deformation [49]. Considering the intersections of slip band as the potential nucleation site of martensite, the kinetics of strain-induced martensitic transformation can be described as follows [49,94]:(5)$$f_{\alpha^{\prime }}=1-\exp\left[-\beta \left(1-\exp{\left(-\alpha \varepsilon \right)}^{2}\right)\right]$$
where *f_α’_* is the volume fraction of transformed martensite and *α* and *β* are the martensitic nucleation rate and the possibility of a martensite embryo at the shear bands, respectively. Martensitic transformation is a nucleation dominated process, thus the nucleation site or embryo created by plastic deformation is important for the formation of martensite during plastic deformation. Nevertheless, it should be noted that the strain-induced martensitic transformation also involves the applied stress that contributes to the martensitic transformation in terms of mechanical interaction energy [92,95]. It is expected that the stress (*σ_b_*) required to induce the martensitic transformation during plastic deformation could be lowered as compared with the critical stress (*σ_c_*) for stress-induced martensitic transformation (Figure 5a), which may be considered as the specific contribution from the creation of the nucleation site by plastic deformation. Nevertheless, the exact role of applied stress on the growth of the martensite embryo created by the plastic deformation needs to be clarified.

The partitioning of stress/strain onto the austenite grains depends on the relative strength of austenite as compared with the adjacent phases [96,97]. In general, the strength of austenite should be higher than the ferrite but may be lower than the bainite and martensite. It is reported that the strain partitioned onto the ferrite is slightly larger than the austenite during Lüders band deformation in a medium Mn steel [98,99]. The blocky austenite is partitioned with more strain as compared with the bainite and martensite in carbide-free bainitic steel [100]. The strain partitioning among the constituting phases is found to be important in influencing the austenite stability in the TRIP steel with Al alloying [101] and the medium Mn steel [79]. Nevertheless, the stress partitioning is reported to play an inferior role in inducing the different austenite stability among the austenite grains adjacent to the bainitic ferrite as compared with the one close to the polygonal ferrite [102]. The difference of localized C content is found to be more important for the observed varied austenite stability among the austenite grains with different adjacent phases [102]. This is consistent with a recent study on TRIP-assisted multiphase steel that the austenite grains with a low C content transform earlier, resulting in more stress partitioning onto the remaining austenite grains with a high C content [103].

#### 2.2.2. Stress State

The service of AHSS in different industrial sectors involves the complex stress state that is beyond the uniaxial stress state provided by the uniaxial test. Therefore, it is vital to identify the response of austenite to the stress state. The martensitic transformation in austenite grains accompanies a transformation strain consisting of shear and dilatational components. The applied stress with varied state interacts with the different components of transformation strain, leading to the different amount of mechanical interaction energy [87]. The mechanical interaction energy can be algebraically incorporated in the change of chemical free energy to account for the effect of stress state on the austenite stability. The mechanical interaction energy (*G_mech_*) can be described as follows [87]:(6)$$G_{mech}=\frac{1}{2}\gamma_{0}\sigma_{1}\sin2\theta \pm \frac{1}{2}\varepsilon_{0}\sigma_{1}\left(1+\cos2\theta \right)$$
where *γ*_0_ is the shear component, *ε*_0_ is the dilatational component, *σ*_1_ is the applied uniaxial stress, and *θ* describes the angle between the applied stress and the normal direction of the martensite plate. For a uniaxial tensile test, a simple relation between the mechanical interaction energy and uniaxial tensile stress can be described as Δ*G_mech_* = 0.86*σ* [104].

The martensitic transformation is facilitated by the uniaxial tensile stress and is less aided by the uniaxial compressive stress (Figure 6) [87]. This is because the uniaxial compressive stress can be resolved into the stress components, where the normal stress opposes the transformation while the shear stress facilitates the transformation [87]. The martensitic transformation is suppressed by the hydrostatic pressure because the hydrostatic pressure opposes the expansion of martensitic transformation (Figure 6) [105]. The increased *M_s_* temperature induced by uniaxial stress state as predicted by the Patel–Cohen model is highly consistent with the experimental observation. However, the prediction of the effect of hydrostatic pressure on *M_s_* temperature yields with a ratio of −5.5 °C for every 100 MPa, which is slightly lower than the experiments [87]. Besides, the linear relation between the decreased *M_s_* temperature and the applied hydrostatic pressure breaks when it is higher than 750 MPa for Fe–Ni alloy [106], which could be owing to the Invar effect on volume expansion of martensitic transformation under hydrostatic pressure [107]. A tensile hydrostatic pressure could be generated on the austenite grains embedded in the ferritic matrix owing to the different thermal expansion ratio and may facilitate the martensitic transformation [28]. An analytical model is derived to explain the change of *M_s_* temperature under the complex stress state [108]. It is found that the change of *M_s_* temperature is mainly affected by the maximum and minimum principal stress and is independent of the intermediate principal stress [108]. A more sophisticated self-consistent model by considering the orientation relationship is developed to evaluate the effect of the stress state on the austenite stability [109].

The shear stress state enhances the martensitic transformation over a wider deformation regime as compared with the uniaxial tensile stress in a TRIP-assisted steel [110]. However, the underlying mechanism for such a difference is not clarified in [110] owing to the complex interplay of different governing factors such as orientation and load partitioning. It is found that the amount of transformed martensite during biaxial tension is two times larger than the uniaxial tension under the same effective strain, which may be owing to the higher amount of intersections of the shear band (nucleation site of martensite) formed during biaxial tension [111,112]. Therefore, the austenite grains are less stable during biaxial tension in comparison with the uniaxial tension.

#### 2.2.3. Strength of the Matrix

The effect of matrix strength on the austenite stability is correlated to the partitioning of stress/strain on austenite grains during the tensile test. The higher strength of the matrix indicates the lower amount of partitioned stress/strain on the austenite grains. Nevertheless, the strength of the adjacent phase affects the austenite stability not only through the stress/strain partitioning, but also through the plastic accommodation for the transformation strain [113]. The austenite grains in AHSS can be surrounded by varied ferritic phases such as the ferrite, bainite (granular/upper/lower), and martensite (fresh/tempered), resulting from the complex phase transformations during the thermal-mechanical process. The strength of these ferritic phases could be sharply different owing to the varied substructures yielded from the diverse transformation mechanisms (diffusional/diffusionless). It is found that the ferritic phases in ascending order of nanohardness are ferrite, granular bainite, upper bainite, lower bainite, tempered martensite, and fresh martensite (Figure 7) [65,114].

The effect of strength of the adjacent phase on austenite stability is mainly investigated by experiments [86,89,115]. The austenite grain in adjacent to the martensite is more stable than the austenite grains close to the ferrite grains in a TRIP-assisted multi-phase steel, which is explained by the less partitioned stress of austenite owing to the adjacent high strength martensite [86]. Similarly, an application of the tempering process on bainitic steel reduces the strength of bainite by decreasing the dislocation density, destabilizing the austenite as compared with the one without tempering [115]. The underlying reason is ascribed to the reduced strength of bainite so that adjacent austenite is partitioned with higher stress [115]. Besides, the tempering process could also lead to a reduced compressive stress field, which may facilitate the martensitic transformation [115]. For the same reason, the austenite grains in the martensite matrix after tempering are less stable owing to the reduced strength of martensite [57,116].

The retained austenite grain positioned at the ferrite grain boundaries transforms earlier than the one at the ferrite grain interiors, while both are less stable as compared with the austenite among the martensite laths [117]. The position-dependent austenite stability for the adjacent ferrite can be rationalized by the fact that the initial plastic flow of polycrystalline materials is accommodated by the formation of GND at the grain boundaries to accommodate the misfit of strain [56]. Thus, the austenite grain located at the ferrite grain boundaries is subjected to more deformation as compared with the one at the ferrite grain interiors. Nevertheless, it seems that the shielding effect of martensite owing to its high yield strength dominates over the position-dependent austenite stability. As the strength of bainite is related to its morphology (i.e., granular bainite is weaker than lath bainite), the dependence of austenite stability on the bainite morphology can be well expected [118]. The delta ferrite is frequently found in the medium Mn steel containing the Al content higher than 3 wt.%. Therefore, the austenite grains adjacent to the delta ferrite should be more stable than the ones close to the eutectoid ferrite owing to the higher strength of delta ferrite.

The effect of free surface on the austenite stability is a special case for the effect of the strength of the matrix. The presence of free surface suggests that no adjacent phase opposes the martensitic transformation. In other words, the free surface means that the strength of the matrix is zero. The influence of the free surface on austenite stability can be investigated by observing the martensitic transformation after partially removing the matrix adjacent to the austenite grain [119]. A free surface is always present during the preparation of a sample for microscopy observation. The formation of martensite near the free surface can be termed as the surface martensite, which is different from the martensite transformed in bulk materials [120]. The presence of free surface reduces the strain energy barrier for martensitic transformation, and thus tends to destabilize the austenite grains [121].

#### 2.2.4. Orientation

The influence of orientation on the austenite stability is correlated to the loading direction, thus it could extrinsically affect the austenite stability. In other words, the orientation can be considered as the extrinsic factor governing the austenite stability. The dependence of austenite stability on the orientation can be observed during the plastic deformation, where a certain orientation for martensitic transformation is favored [122,123,124,125]. The austenite grains with the preferred orientation transform earlier owing to higher resolved shear stress [123]. In other words, the dependence of austenite stability on orientation can be ascribed to the mechanical driving force, which is also orientation-dependent [126,127]. Therefore, the austenite grains with an orientation that gain higher mechanical interaction energy are less stable during the plastic deformation. In general, the plastic deformation of austenite can be accommodated by grain rotation and multiple plasticity mechanisms (dislocations, martensitic transformation, and deformation twins). The grain rotation is beneficial to reduce the misfit resulted from the varied Schmidt factors of grains during initial plastic deformation [56]. Thus, it is expected that the austenite grains with low stability could transform to martensite before rotation, while the austenite grains with high stability tend to rotate to accommodate the initial plastic deformation [128,129]. It is reported that the rotation of austenite grains decreases the Schmidt factors (resolved shear stress), and thus reduces the mechanical interaction energy for martensitic transformation [129]. The influence of orientation on the austenite stability can be modeled through the stress-induced martensitic transformation [126,127] or by the phenomenological theory of martensite crystallography (PTMC) [130]. However, the effect of orientation plays a secondary role in affecting the austenite stability as compared with the grain size [128] and chemical compositions [122,123].

The effect of initial orientation on the austenite stability is difficult to evaluate through macroscopic plastic deformation owing to the rotation of austenite during the deformation process. Nevertheless, it may be clarified through the deformation of a single crystal with varied orientation [131]. The effect of orientation on the martensitic transformation can be manifested by critical stress for formation of martensite in a single crystal during the compression test [132]. It is found that the critical stress for initiating martensitic transformation in the [100] micropillars is around 200 MPa lower than that in the [110] micropillars [132]. The underlying mechanism for the different dependence of austenite stability on orientations is ascribed to the stress required to induce the deformation twins. The deformation twins are only found in the [100] micropillars. The intersection of the deformation twins may serve as the nucleation site of martensite, and thus lower the critical stress for martensitic transformation in [100] micropillars [132].

#### 2.2.5. Strain Rate

The design of the high-performance of AHSS needs to consider the mechanical behaviors at a high strain rate because the sheet metal forming and collision during crash events involve high strain rate deformation (200–500 s^−1^). As the mechanical behaviors of AHSS relies on austenite stability, it is essential to understand the influence of strain rate on the austenite stability. In general, two aspects should be considered for the effect of strain rate on the austenite stability, including the nucleation of martensite and the adiabatic heating. It is expected that the heat dissipation resulting from 90% of the plastic work drives the rise in temperature of samples at a high strain rate [133]. The plastic work is proportional to the plastic strain. Therefore, the adiabatic heating is limited at a low strain level, while it is significant at a high strain level [134].

For metastable austenitic stainless steel, it is found that the austenite grain is less stable at a high strain rate (10^3^ s^−1^) under a strain lower than 0.25 on the aspect that the amount of transformed martensite at high strain rate is larger as compared with the one at a low strain rate (10^−3^ s^−1^) (Figure 8) [112]. This can be ascribed to the increased amount of shear bands, where the intersections of these shear bands could become a potential nucleation site, and thus leads to the larger martensite volume fraction at a high strain rate under a strain smaller than 0.25 [111]. However, the temperature increase at a high strain rate (10^3^ s^−1^) under a high strain level (>0.25) owing to the adiabatic heating considerably suppresses the martensitic transformation and leads to the reduced amount of transformed martensite (Figure 8) [112,135]. The above finding is correlated to the complex strain sensitivity in the stainless steel. The austenitic stainless steel has positive strain rate sensitivity at a low strain level (<0.2), while it has negative strain rate sensitivity under a high strain level [136,137]. The positive strain rate sensitivity means that the high strain rate leads to higher flow stress, and vice versa for the negative strain rate sensitivity. The tensile test at a high strain rate under a low strain level (<0.2) leads to the formation of a larger amount of martensite, which improves the work hardening behaviors, and thus demonstrates the positive strain rate sensitivity [136,137]. The suppression of the martensitic transformation at a high strain rate under a large strain level (>0.2) reduces the work hardening behaviors, and thus shows negative strain rate sensitivity.

The influence of strain rate on the austenite stability is more complex for multi-phase steel. Irrespective of the strain level, it is found that the amount of transformed martensite under an ultralow strain rate (~10^−6^ s^−1^) and a high strain rate (~2 × 10^2^ s^−1^) is lower than the one under a medium strain rate (~2 × 10^−3^ s^−1^) for a typical medium Mn steel, which is rationalized by considering the different strain partitioning between the austenite grains and adjacent phase at the different strain rate [139]. It is found that the TRIP-assisted multi-phase steels with needle-like austenite grains demonstrate better mechanical properties at a high strain rate test as compared with the counterpart with blocky austenite grains [140]. A non-monotonic strain rate effect on austenite stability, that is, the stabilization of austenite at a low strain rate (2 × 10^−4^ − 10^−1^ s^−1^) owing to thermal effect (adiabatic heating) and acceleration of martensitic transformation at a high strain rate (0.1 − 175 s^−1^) owing to the generation of more martensite nucleation sites, is reported in Q&P 980 steel [141]. The above explanation could be reasonable by considering the possibility that the adiabatic heating in Q&P 980 steel at the high strain rate of 175 s^−1^ is not significant because the total elongation is less than 10% [111,141]. The high strain rate of 10^3^ s^−1^ reduces the austenite stability in the Q&P 1500 steel at the low strain level (<0.04) (Figure 8) [138]. Despite enhanced martensitic transformation at a high strain rate, the corresponding strain hardening is substantially decreased, which is ascribed to the negative strain rate sensitivity of the martensite matrix and the abnormal TRIP effect [138]. The abnormal TRIP effect is ascribed to the plastic deformation of transformed martensite at a high strain rate [138].

As the adiabatic heating is inevitable during crash events, it may be desirable to reduce the austenite stability so that a large proportion of the austenite grains could transform to martensite at the strain where the adiabatic heating is not significant. Then, the operation of composite effect among the ferritic matrix and transformed martensite enhances the strain hardening at the high strain level. Note that most of the high strain rate tests are not able to approach the speed of martensitic transformation, that is, the transmission speed of sound in solids [142]. It is interesting to identify the response of austenite at an extremely high strain rate, that is, approaching or beyond the speed of martensitic transformation. Such a high strain rate can be realized in the spallation process or in the high-speed machining process (~10^7^ s^−1^).

## 3. Characterization Techniques

The martensitic transformation in the austenite grains can be induced by deformation [87], continuous cooling [59], magnetic force [143], or a combination of the above processes. The effect of different factors including intrinsic and extrinsic factors on the austenite stability can be determined by different techniques, including the tensile test, nanoindentation test, dilatometry test, magnetization test, and in situ observations.

### 3.1. Tensile Test

The uniaxial tensile test is one of the most basic techniques to evaluate the mechanical properties of structural materials. Through such a test, the strength (yielding/tensile) and ductility (uniform/total) can be directly obtained (Figure 9). The evolution of microstructure during uniaxial tension can be captured by detailed characterization on the tensile samples stopped at varied strains [20,63]. For instance, the martensitic transformation kinetics and the evolution of dislocation density during uniaxial tension can be obtained through the X-ray diffraction (XRD) test. The tensile testing armed with the digital image correlation (DIC) technique is effective in evaluating the strain partitioning among the phases and the corresponding effect on austenite stability in steels [144].

The stress partitioning among phases during the tensile testing can be obtained by employing the neutron diffraction analysis [144]. By incorporating the environmental box, the effect of temperature on the austenite stability can be investigated [145]. The tensile testing generally reveals the overall effect of intrinsic and extrinsic effects on the stability of austenite [89,146,147]. In other words, the austenite stability measured during the tensile test is affected by both intrinsic and extrinsic factors. The factors that intrinsically affect the austenite stability during the tensile test include the chemical composition, morphology, size, and dislocation density [81]. The extrinsic factors mainly include the stress/strain partitioning, the strength of the matrix, and the local orientation [128]. Therefore, it is difficult to identify the influence of individual factors on the austenite stability through the tensile test as these factors in most cases are coupled during plastic deformation. For instance, the influence of morphology on the austenite stability in a Q&P steel during uniaxial tension involves the effect of C content and the strength of adjacent phases [22].

### 3.2. Nanoindentation Tests

Different from the macroscopic tensile test, the nanoindentation test armed with a sharp indenter is effective in obtaining the mechanical stability of single austenite grains [148,149]. The formation of martensite beneath the indenter may contract along the indentation loading direction, generating a strain burst or pop-in in the load–displacement (*P*–*h*) curve (Figure 10) [148]. Nevertheless, not all the pop-ins shown in the *P*–*h* curve represents the formation of martensite beneath the indenter. By fitting the elastic part of the *P*–*h* curve using the Hertzian solution, it is generally found that deviation of the *P*–*h* curve from the Hertzian solution is accompanied by a pop-in (Figure 10), which could represent the incipient plasticity or nucleation of dislocation [150]. The second pop-in could be induced by the formation of martensite beneath the indenter because the deformation mode is changed [149,151]. Therefore, the load corresponding to the occurrence of the second pop-in may represent the critical load to initiate the formation of martensite in the single austenite grains. The critical load is found to increases with the Mn content, confirming that the Mn element is an effective austenite stabilizer [149]. Consequently, the effect of chemical compositions such as Mn content on the austenite stability can be examined by the nanoindentation test. It is found that the grain refinement suppresses the occurrence of strain burst in metastable stainless steel, suggesting that the grain refinement contributes to the austenite stability [152]. The influence of the prior martensite on the austenite stability has been investigated by nanoindentation [153]. It is found that the prior martensite stabilizes the remaining austenite grains by the hydrostatic pressure and the dislocations, both of which result from the formation of adjacent martensite [153,154]. Therefore, it is believed that the nanoindentation test is a useful technique to determine the intrinsic effect on the austenite stability as the indented volume is generally confined within an individual austenite grain.

The nanoindentation test with a Berkovich indenter involves the generation of GND loops beneath the indenter, leading to the development of a very complex stress field. Therefore, only the critical load rather than the critical stress is employed to evaluate the austenite stability for the nanoindentation test with a Berkovich indenter [149]. However, the uniaxial compression test on a pillar made in an austenite grain using a flat punch can induce a homogeneous stress state without the influence of GND [155]. Similarly to the nanoindentation test on bulk austenite using a Berkovich indenter, the nanoindentation compression test with a plat punch on a single crystal of austenite as fabricated by the focus ion beam (FIB) milling also leads to the occurrence of strain burst [131,156]. The strain burst is explained by the shear transformation of martensite along the compression direction [156]. The benefit of such measurement is that the influence of orientation on the austenite stability can be clearly clarified [132]. Nevertheless, the compression test on the micropillars generally involves the single austenite crystal with a large surface to volume ratio, which may affect the martensitic transformation behavior [156].

### 3.3. Dilatometry Test

The high-precision dilatometer is an ideal technique to measure the dilatation of phase transformation during thermal processing (Figure 11). Although the shape strain of martensitic transformation is dominated by shear, the shear strain may be cancelled out owing to the nucleation of martensite with different variants [157]. Thus, it is reasonable to expect that the dilatation measured by a dilatometer push rod represents the volume expansion of martensite in the cylinder sample. Based on an isotropic assumption, the dilatation obtained with respect to the temperature is capable of calculating the *M_s_* temperature and the transformation kinetics. Thus, the dilatometer has been extensively used to investigate the phase transformation in steels (Figure 11) [158,159,160]. The influence of grain size on the austenite stability has been investigated by dilatometer based on the measurement of the *M_s_* temperature under varied austenitization temperature and time (varied austenite grain size) [61]. Besides the grain size, the deformation-dilatometer allows the investigation of the effect of applied stress on the austenite stability [161]. It is generally found that the applied stress below the yield strength assists the martensitic transformation in that the *M_s_* temperature is increased with an application of applied stress, which can be rationalized based on the concept of mechanical interaction energy.

The influence of dislocations (ausforming process) on the austenite stability can be studied by deformation-dilatometer [162]. The dislocations are introduced by plastic deformation without austenite decomposition, which is also known as the ausforming process. The austenite after deformation is subjected to continuous cooling to capture the following martensitic transformation kinetics [52]. Both intrinsic (chemical compositions, grain size, and dislocations) and extrinsic factors (applied stress) on the austenite stability can be investigated by employing the dilatometer. Nevertheless, it is not clear whether the factors beyond the above examples can be investigated by the dilatometer measurement.

### 3.4. Magnetization Test

The transformation from paramagnetic austenite to ferromagnetic martensite can be affected by an application of a magnetic field [143,163,164]. The influence of the magnetic field on the martensitic transformation is firstly detected by Herbert, who discovered an enhanced hardness of martensite with the application of the magnetic field in 1929 [165]. Later, the above phenomenon is explained by Sadovsky et al., who found that the *M_s_* temperature in an Fe-Ni-Cr-C alloy is increased with the application of a magnetic field [166,167]. Moreover, the amount of transformed martensite is also increased after an application of the magnetic field [164,168]. Besides, the incubation time of martensitic transformation in stainless steel is reduced by the magnetic field [169]. All of these evidences, including the elevated *M_s_* temperature, increased amount of transformed martensite, and reduced incubation time, indicates that the martensitic transformation is accelerated by the magnetic field [164]. However, it seems that there is a critical magnitude of the magnetic field below which no martensitic transformation is observed [170]. Interestingly, the volume fraction of transformed martensite is independent of the peak strength of the magnetic field once the critical magnitude of the magnetic field is achieved, which is ascribed to the burst characteristic of martensitic transformation in steels [171]. The acceleration of martensitic transformation by a critical magnetic field can be understood from the aspect of Zeeman energy, high field susceptibility energy, and forced volume magnetostriction energy [171]. The thorough reviews on the influence of the magnetic field on the martensitic transformation in steels can be found in [107,171,172].

As the martensitic transformation can be driven by a magnetic field at a constant temperature, it can be used to investigate the effect of intrinsic factors on austenite stability without resorting to the cooling or deformation processes. This can disregard either the thermal stress among phases during the cooling process or the stress/strain partitioning during uniaxial tension. In other words, the magnetic field is ideal to investigate the intrinsic factors governing the austenite stability.

### 3.5. In Situ Techniques

The benefit of in situ observation on the austenite stability is that the formation of martensite with respect to the specific strain or temperature can be clearly identified. The in situ observation can be carried out under optical microscope (OM) [173], scanning electron microscope (SEM) [79], transmission electron microscope (TEM) [174,175], and synchrotron X-ray diffraction (S-XRD) [28,176]. Note that the in situ observation itself does not induce the martensitic transformation in metastable austenite grains, which should be accompanied by an application of cooling or deformation processes to drive the martensitic transformation.

Although the in situ observation can obtain direct evidence for the factors governing the austenite stability, it frequently involves the effect of the free surface as the observation is mainly performed on the surface. It is found that the amount of martensite transformed close to the free surface is larger than the bulk interior owing to the lack of constraint from adjacent grains [121]. In particular, it is expected that the influence of free surface on the austenite stability is severe under in situ TEM observation, as the thickness of the TEM sample should be as thin as around 100 nm to allow the transmission of electron, and such a thickness is even smaller than the width of lath martensite (~100−500 nm) [65]. In other words, the observation of the martensitic transformation in metastable austenite grains using microscopy such as the TEM observation may undesirably involve the effect of the free surface [129,174,175]. Therefore, it is not clear whether the in situ microscopy observation can represent the transformation behavior in its bulk counterpart. In contrast, the information obtained by the synchrotron test is around 20 μm beneath the surface and may represent the bulk transformation behavior considering the ultrafine austenite grains’ size in most of the AHSS [177]. The neutron test can fully capture the bulk behavior owing to its high transmission capacity [178]. However, the samples after the neutron test must be left for a few years to allow additional microscopic observation owing to safety issues related to radiation.

S-XRD has been employed to investigate the stability of single austenite grains in the TRIP steels [58,59,60]. The austenite volume fraction during the cooling process can be estimated from the diffracted intensity of phases. The C content of austenite grains can be assessed from the lattice parameter acquired by the in situ S-XRD test [28]. Interestingly, the C content of austenite in Q&P 980 steel is found to be different with respect to the austenite morphology, which is concluded from the observation of two sub-peaks from a single austenite diffraction peak [22]. The influence of grain size on the austenite stability can be identified through the in situ S-XRD test [58,59,60]. However, the thermal stress resulting from the different expansion ratios of phases may facilitate the martensitic transformation in such an observation during the cooling process [28]. By combining the S-XRD and tensile test, the effect of orientation on the austenite stability can be clarified [122]. Besides the effect of orientation, the loading partitioning and its effect on the austenite stability can be obtained [110]. The stress partitioning among phases and its influence on the austenite stability in TRIP-assisted steels can be studied by the synchrotron/neutron diffraction [103,124,179]. The effect of strength of the bainitic matrix through varied dislocation density on the stability of austenite has been evaluated by S-XRD [115].

Therefore, both intrinsic (chemical compositions, grain size, and morphology) and extrinsic factors (stress partitioning, orientation, and the strength of the matrix) on the austenite stability can be investigated by employing in situ techniques, especially S-XRD and neutron diffraction. Nevertheless, it is not clear whether the other factors such as strain rate can be investigated by in situ S-XRD.

## 4. Alloy Design Strategies

It is generally believed that a large austenite volume fraction with optimized stability is desirable for developing strong and ductile steel for automotive application [180,181], which can be achieved by controlling the intrinsic factors (chemical compositions, dislocations, grain size, and morphology) through thermal-mechanical processing. The chemical compositions play a pivotal role in controlling the austenite stability at room temperature, which is assisted by other intrinsic factors. The heat treatment of the third AHSS is designed to allow partitioning of C and/or Mn to stabilize austenite grain at room temperature. The intrinsic factors that govern the austenite stability in medium Mn TRIP steel can be tuned based on the annealing temperature [182], duration [183], deformation (warm rolling/cold rolling) [184], and initial starting microstructures [185]. In addition to the conventional C and Mn partitioning for medium Mn TRIP steel during the intercritical annealing [19,20], the Mn segregation at different length scales [186,187,188,189,190,191,192] can also be employed to stabilize the austenite grains. The ultrafine austenite grain size is another vital factor in governing the austenite stability in medium Mn steel [63,193,194]. The Q&P process employed for low alloyed steel generates a dual-phase martensite matrix and refined austenite grains, where the former partitions C into the latter [21,195]. The bainite transformation at a low temperature not only allows the C partitioning into the adjacent austenite grains, but also substantially refines the austenite grains [23,196]. Note that the austenite stability in Q&P steel and CFB steel could be contributed by the hydrostatic pressure and dislocations, both of which are correlated to the formation of martensite/bainite. The austenite stability can be reset by engineering dislocations using a warm rolling process [197,198,199]. The austenite stability can be improved with grain refinement achieved through the near Ac3 temperature annealing [200], thermal cycling [201], and nano-precipitations [202,203]. The morphology of the austenite grains can be optimized by tuning the annealing temperature to enhance the performance of nanostructured TRIP-assisted steels [204].

The austenite stability is intrinsically governed by the varied factors within single austenite grains in opposing the martensitic transformation triggered by an external stimulus such as deformation or cooling processes. Therefore, besides the intrinsic factors, the extrinsic factors should also be considered in controlling the austenite stability. The partitioned stress/strain is a vital extrinsic factor in affecting the austenite stability and is correlated to the strength of adjacent phases. If the partitioned stress does not effectively increase the mechanical interaction energy to help the austenite grains overcome the nucleation barrier, the TRIP effect resulting from the martensitic transformation may be suppressed during the plastic deformation [205]. In this case, the improvement of work hardening behaviors assisted by the TRIP effect is missed, leading to the relatively low work hardening rate. Either the relaxation of matrix strength so that more stress can be partitioned onto the austenite or the reduction of the intrinsic austenite stability can be employed to realize the potential of the TRIP effect to improve the ductility [97,206]. It has been reported that the orientation of the austenite grains can be utilized to tailor the austenite stability to enhance the ductility of steels [129]. To enable the full potential of the TRIP effect, the intrinsic austenite stability should be balanced with extrinsic austenite stability.

Although the governing factors on the austenite stability can be separated into intrinsic and extrinsic components, both intrinsic and extrinsic factors could be controlled by the same microstructure, such as grain size and dislocations. The yield strength of metallic materials is determined by the grain size through the Hall–Patch relation [4,5] or is affected by the dislocation density through the Taylor hardening [7]. The increased yield stress may elevate the stress that is partitioned on the austenite grains. The enhanced austenite stability can be achieved through grain refinement and high dislocation density. Thus, the competition between the intrinsic and extrinsic factors induced by the same microstructural features may lead to the simultaneous increase of strength and ductility. In other words, by enabling the synergetic effect of defects in controlling the austenite stability with consideration of the balance between the intrinsic and extrinsic factors, the optimized TRIP effect may be able to defeat the strength–ductility trade-off, which has been demonstrated in the austenitic stainless steel [175] and the high entropy steel containing metastable austenite phase [207]. The yield strength of high entropy steel is improved after the grain refinement. The reduced grain size also stabilizes the austenite (face centered cubic, fcc) grains, reducing the transformation rate to provide the controlled release of the TRIP effect (Figure 12) [207]. However, the scenario for dislocations could be slightly different. As discussed in previous paragraphs, the influence of dislocations on the austenite stability depends on the amount of deformation. It is found that the warm rolling of AISI 301 stainless steel with a thickness reduction of 20% facilitates the martensitic transformation, enabling an enhanced TRIP effect to improve the work hardening behaviors (Figure 12) [175]. However, the austenite grains after the large warm rolling reduction of 40% could have too many dislocations, and thus are too stable to transform, although the applied stress is elevated owing to the introduction of high dislocation density [174]. The alloy design strategy, by utilizing the competing role of grain size and the complex role of dislocations in affecting the austenite stability, can be worthwhile as it may lead to the development of lean, high-performance steel for automotive applications.

## 5. Prospect

The austenite stability in steels is governed by intrinsic and extrinsic factors. The deformation twins, which are observed in the austenite grains with proper stacking fault energy [208], can be classified as the intrinsic factors affecting the austenite stability. However, the influence of the deformation twins on the austenite stability is not clear yet. It is reported that the intersection of two deformation twins acts as the nucleation site of martensite [49,208,209]. From this aspect, the deformation twins are expected to facilitate the nucleation of martensite. However, the deformation twins may inhibit the growth of the martensite by acting as barriers for the glissile austenite/martensite interface, which is similar to the mechanical stabilization of austenite controlled by the dislocations [51]. It has been postulated that the growth of martensite could be advanced by generation and coalescence of new embryos [111]. In this case, the nucleation rather than the growth dominates the martensitic transformation kinetics. However, the detailed mechanism of the effect of the deformation twins on the austenite stability should be substantiated by the systematic experiments and modeling works.

The combined effect of intrinsic and extrinsic factors on the austenite stability is systematically evaluated by experiments on different steel grades. However, the corresponding modeling work is still lacking in general. By incorporating the influence of chemical composition and deformation temperature in terms of free energy on the martensitic transformation, the evolution of austenite volume fraction to plastic strain can be predicted by a mathematical model [210]. Nevertheless, the other governing factors such as the morphology and stress partitioning are difficult to incorporate in this mathematical model. The numerical method that considers the geometry of grains may be useful to capture the effect of varied factors on the austenite stability. The phase-field modeling may advance the understanding of the growth of martensite [121,211]. The crystal plasticity finite element method (CPFEM) has been demonstrated to be capable of investigating the influence of austenite kinematic stability on the transformation and deformation of duplex stainless steel [212]. It is believed that the micromechanical modeling or the CPFEM may be helpful to identify the effect of austenite stability on the mechanical behavior of the third AHSS, although the development of an appropriate constitutive law in capturing the austenite stability is generally challenging currently. The integration of phase-field modeling and CPFEM may bring insights into the understanding of the effect of different factors on the austenite stability.

The present review separates the governing factors on austenite stability into intrinsic and extrinsic factors, which successfully rationalizes the observation that, the stronger, the more ductile in certain alloys with grain refinement and high dislocation density. In other words, the competition between the intrinsic factors and extrinsic factors in affecting the austenite stability can be utilized to overcome the strength–ductility trade-off in steels containing metastable austenite grains. Although a few experimental works have been performed to use the synergy of dislocations and grain size in affecting the austenite stability, the application of this novel alloy design strategy should be extended to improve the mechanical properties of the third AHSS. Besides, the modeling work on the synergy of the defects in affecting the austenite stability, and thus the mechanical properties of steels, can provide insights into thermal-mechanical processing to answer the questions such as, what is the optimal dislocation density or grain size to reveal the potential of austenite stability in achieving the best properties for engineering applications. Such an effort is meaningful as it fully utilizes the defects without resorting to the additional alloying, enabling high-performance steels for applications in resource-limited earth.

## 6. Conclusions

The present review separates the factors governing austenite stability into intrinsic and extrinsic factors based on the domain of individual austenite grains. The intrinsic factors include the chemical compositions, dislocations, grain size, and morphology, while the extrinsic factors include the stress/strain partitioning, stress state, strength of the matrix, orientation, and strain rate. The effectiveness of these factors in affecting the austenite stability is discussed and can be evaluated by different techniques (i.e., tensile test, nanoindentation test, dilatometry test, magnetization test, and in situ techniques). A new alloy design strategy that involves the competition between the intrinsic and extrinsic factors through the same microstructural features such as grain size and dislocations in affecting the austenite stability is proposed. It is suggested that the design of the AHSS containing austenite should consider these intrinsic and extrinsic factors to fully optimize the austenite stability to enable the great potential of the TRIP effect in enhancing the mechanical properties.

## Figures and Tables

**Figure 1 materials-13-03440-f001:**
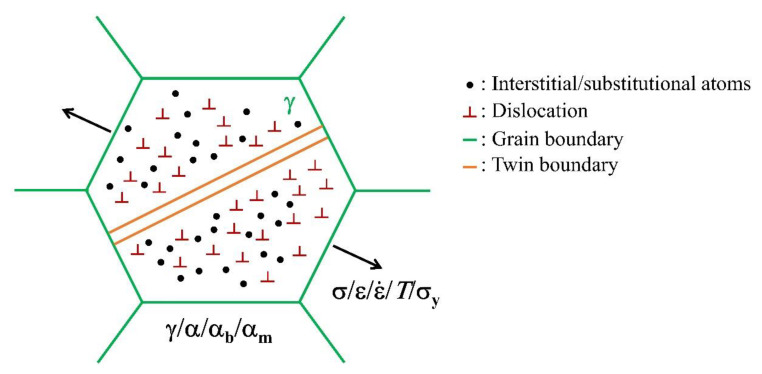
Schematic illustration of the intrinsic and extrinsic factors in governing the austenite stability. *γ*: austenite; *α*: ferrite; *α_b_*: bainite; *α_m_*: martensite; *σ*: stress; *ε*: strain, $\dot{\varepsilon }$: strain rate; *T*: temperature; *σ_y_*: yield strength.

**Figure 2 materials-13-03440-f002:**
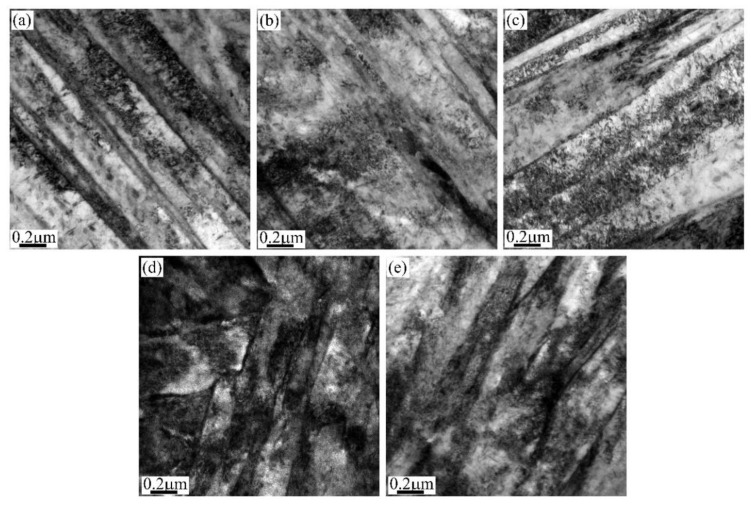
The transmission electron microscope (TEM) images of martensite transformed from the parent austenite in a low C steel (Fe–0.2C–1.5Mn–2Cr in wt.%) with deformation of (**a**) 4.7%, (**b**) 9.4%, (**c**) 18%, (**d**) 25.7%, and (**e**) 36.2%.

**Figure 3 materials-13-03440-f003:**
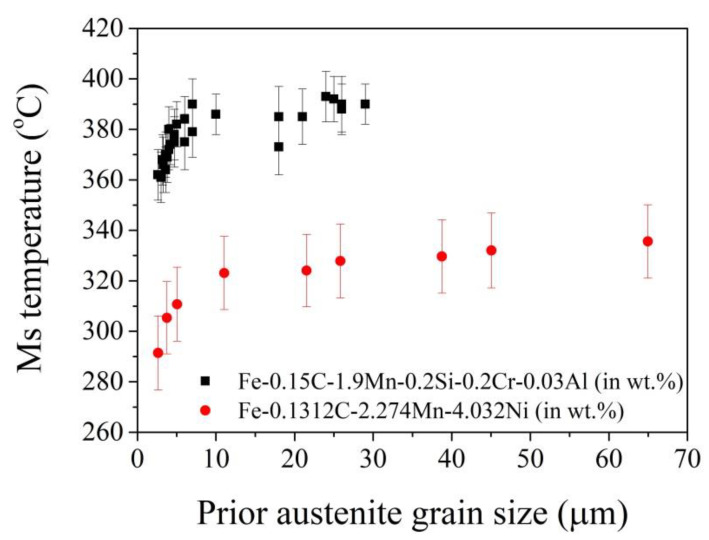
Correlation between prior austenite grain size and *M_s_* temperature in steels with different chemical compositions [61,67].

**Figure 4 materials-13-03440-f004:**
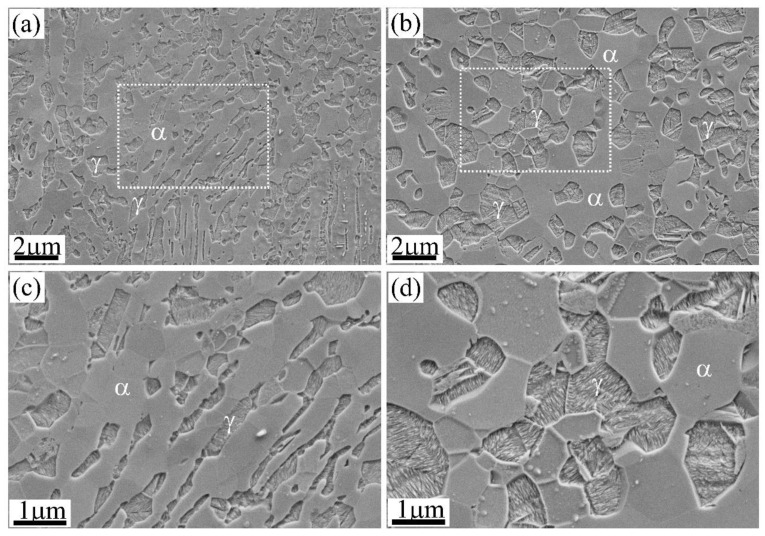
The scanning electron microscope (SEM) images of medium Mn steel with different morphology of austenite resulted from the varied annealing states, including the (**a**,**c**) continuous annealing and (**b**,**d**) batch annealing. γ: austenite; α: ferrite. (**c**,**d**) The enlarged view of dashed rectangles in (**a**,**b**), respectively.

**Figure 5 materials-13-03440-f005:**
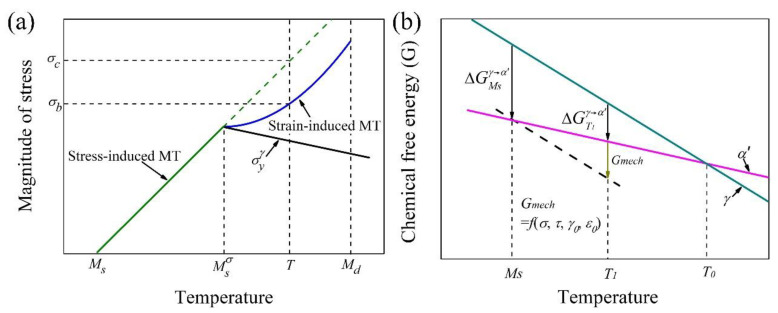
(**a**) Schematic illustration on the definition of stress-induced or strain-induced martensitic transformation (MT) under different deformation temperatures. (**b**) Schematic illustration of the mechanisms of stress-induced martensitic transformation [92].

**Figure 6 materials-13-03440-f006:**
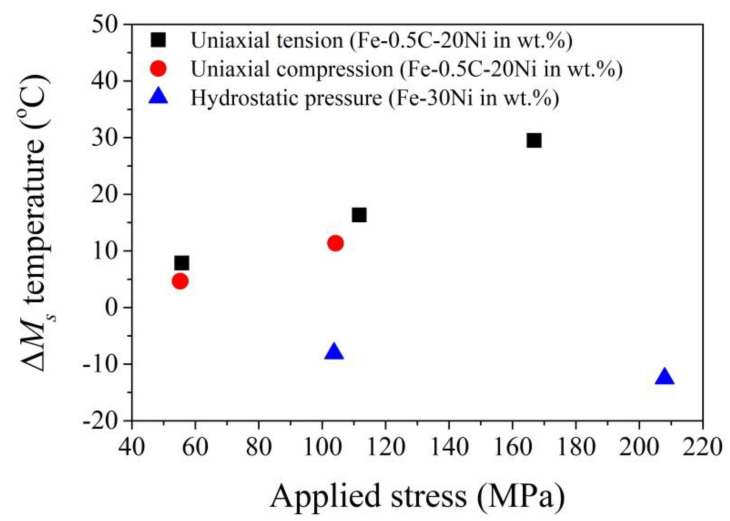
Effect of stress state on the change in *M_s_* temperature in different steel grades [87].

**Figure 7 materials-13-03440-f007:**
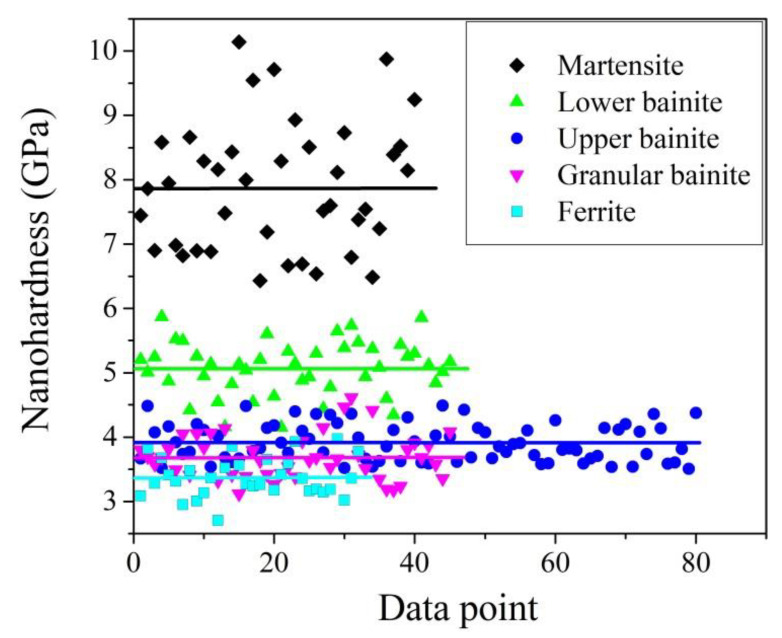
The nanohardness of different ferritic phases in low carbon steel (Fe-0.12C-1.0Si-2.0Mn in wt.%) [114].

**Figure 8 materials-13-03440-f008:**
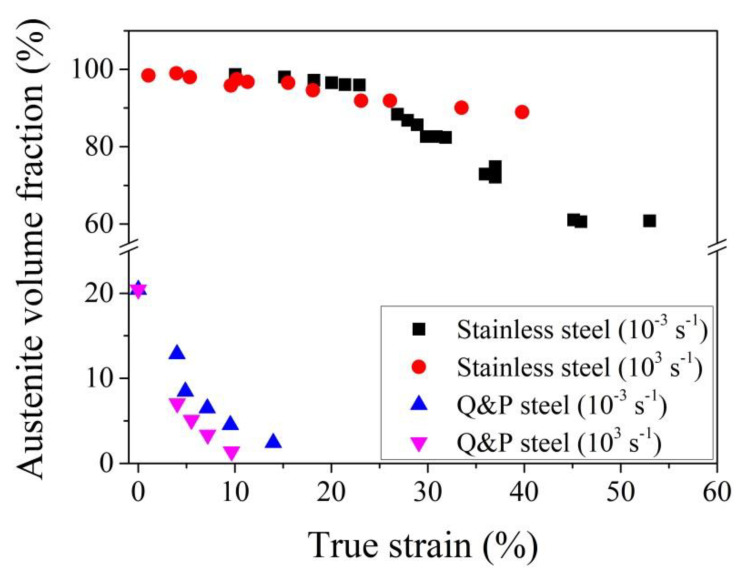
Evolution of austenite volume fraction in stainless steel [112] and quenching and partitioning (Q&P) steel [138] at different strain rates.

**Figure 9 materials-13-03440-f009:**
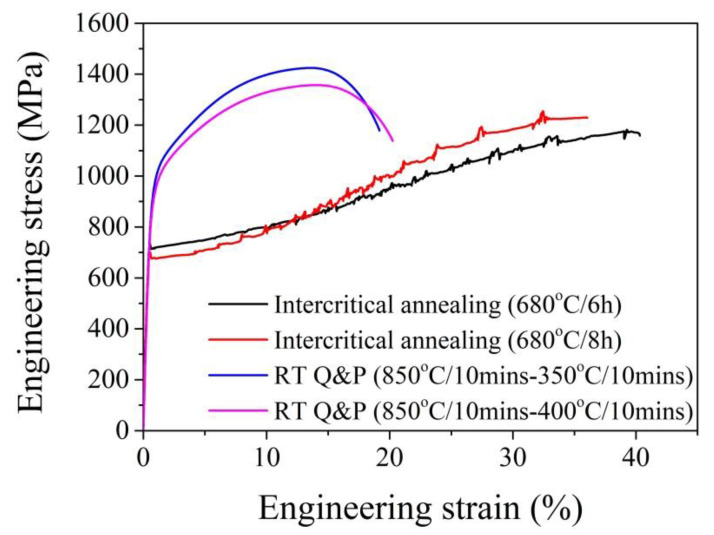
Engineering stress–strain curves of a medium Mn steel (Fe-10Mn-0.2C-2Al-0.1V in wt.%) treated by different thermal processes including intercritical annealing and room temperature quenching and partitioning (RT Q&P).

**Figure 10 materials-13-03440-f010:**
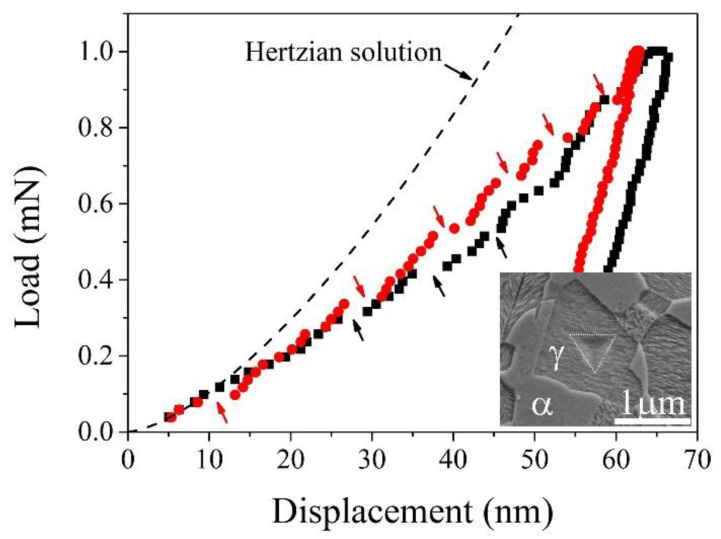
The load–displacement curve of indents on austenite grains in a medium Mn steel (Fe-7.16Mn-0.14C-0.23Si in wt.%). The arrows mark the pop-in. The lower right inset shows the indent on a single austenite grain.

**Figure 11 materials-13-03440-f011:**
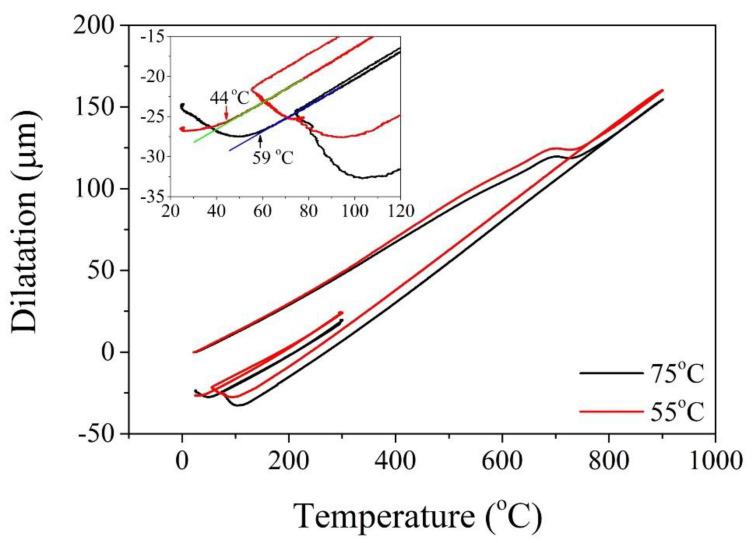
Dilation–temperature curves of medium Mn steel (Fe-8Mn-0.25C in wt.%) subjected to the quenching and partitioning process. The quenching temperature (QT) is either 55 °C or 75 °C, while the partitioning is at 300 °C with a duration of 60 s. The *M_s_* temperature decreases after partitioning in comparison with the quenching temperature, confirming the partitioning of C from martensite to austenite.

**Figure 12 materials-13-03440-f012:**
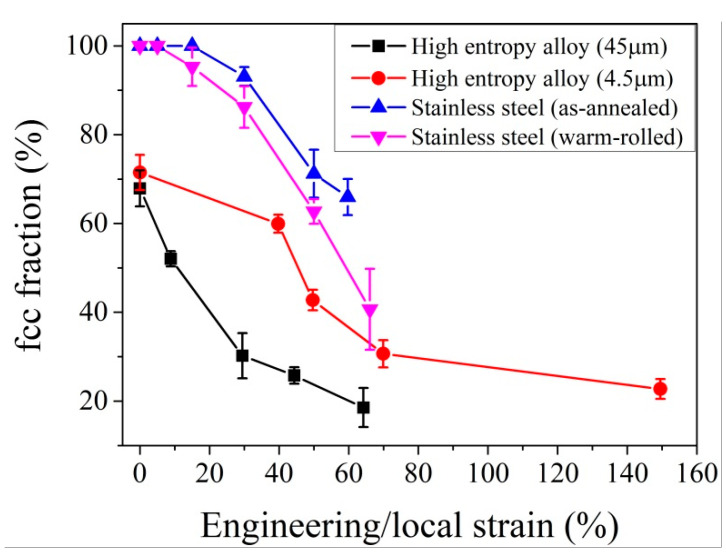
Evolution of austenite (face centered cubic, fcc) volume fraction with respect to strain for different alloys under varied grain size [207] or dislocation density [175].

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
