# Peer review of "On the Factors Governing Austenite Stability: Intrinsic versus Extrinsic"

_materials, 2020, doi:10.3390/ma13153440_

Round 1

Reviewer 1 Report

Errors in the text:

Line 88-89-error description formula

Formula 1 - literature [29], book- the reviewer did not find this formula.  

Line 98 – literature didn’t find equation 2, however in the article: J. TRZASKA CALCULATION OF CRITICAL TEMPERATURES BY EMPIRICAL FORMULAE, Archives of Metallurgy and Materials 2016 | Vol. 61, iss. 2B | 981—986. there is another formula and  article J. Wang, et al. Determination of Martensite Start Temperarure in Engineering Steels. PartI: Empirical Relations Describing in the effect of steel chemistry,  Materials Transactionns, JIM 41, 7, 2000, pp761-768. also gives other formulas, reference should be cited,

Line 181 – formula (4) – line 182 A=423wt%C according to Andrews equation but in the work [55] A=425 wt% (K.W. Andrews, J. Iron Steel Inst. 203 (1965) 721.)

Line 290 error formula (5) description

Line 330- error formula (6) description

Reviewer 2 Report

Reviewers' comments:

Manuscript Number: materials-879026

Full Title: On the factors governing austenite stability: intrinsic versus extrinsic.

Comments: 

- Figure 2. The TEM images of martensite transformed from the parent austenite in a low C steel (Fe–0.2C–1.5Mn–2Cr in wt.%) with deformation of (a) 4.7%, (b) 9.4%, (c) 18%, (d) 25.7%, and (e) 36.2% not clear make clear.

- Figure 12, not clear make clear.

- Check line number 98, 99, 100 and 101.

- Check line number 111, 112, and 117.

- Several faults: are added or missing spaces between words: see manuscript file.

- References: make all references in same format for volume number, page numbers and journal name, because it is difficult to searching and reading.

So that I recommended this manuscript to minor revision and for future process.
